# National Multicenter Study on the Comparison of Robotic and Open Thymectomy for Thymic Neoplasms in Myasthenic Patients: Surgical, Neurological and Oncological Outcomes

**DOI:** 10.3390/cancers16020406

**Published:** 2024-01-18

**Authors:** Elisa Sicolo, Carmelina Cristina Zirafa, Gaetano Romano, Jury Brandolini, Angela De Palma, Stefano Bongiolatti, Filippo Tommaso Gallina, Sara Ricciardi, Michelangelo Maestri, Melania Guida, Riccardo Morganti, Graziana Carleo, Giovanni Mugnaini, Riccardo Tajè, Fabrizia Calabró, Alessandra Lenzini, Federico Davini, Giuseppe Cardillo, Francesco Facciolo, Luca Voltolini, Giuseppe Marulli, Piergiorgio Solli, Franca Melfi

**Affiliations:** 1Minimally Invasive and Robotic Thoracic Surgery—Surgical, Medical, Molecular and Critical Care Pathology Department, University Hospital of Pisa, 56124 Pisa, Italy; carmelina.zirafa@ao-pisa.toscana.it (C.C.Z.); gaetano.romano@ao-pisa.toscana.it (G.R.); fabriziacalabro92@gmail.com (F.C.); alessandralenzini2@gmail.com (A.L.); f.davini@ao-pisa.toscana.it (F.D.); franca.melfi@unipi.it (F.M.); 2Department of Thoracic Surgery, IRCCS Azienda Ospedaliero-Universitaria di Bologna, 40138 Bologna, Italy; jury.brandolini@ausl.bologna.it (J.B.); piergiorgio.solli@ausl.bologna.it (P.S.); 3Unit of Thoracic Surgery, Department of Precision and Regenerative Medicine and Ionian Area, University of Bari “Aldo Moro”, 70121 Bari, Italy; angela.depalma@uniba.it (A.D.P.); g.carleo2@studenti.uniba.it (G.C.); giuseppe.marulli@uniba.it (G.M.); 4Thoracic Surgery Unit, Department of Experimental and Clinical Medicine, Careggi University Hospital, 50134 Florence, Italy; stefanobongiolatti@gmail.com (S.B.); g.mugnaini12@gmail.com (G.M.); luca.voltolini@unifi.it (L.V.); 5Thoracic Surgery Unit IRCCS Regina Elena National Cancer Center, 00144 Rome, Italy; filippo.gallina@ifo.it (F.T.G.); riccardo.taje@ifo.it (R.T.); francesco.facciolo@ifo.it (F.F.); 6Unit of Thoracic Surgery, Azienda Ospedaliera San Camillo Forlanini, 00152 Rome, Italy; ricciardi.sara87@gmail.com (S.R.); gcardillo@scamilloforlanini.rm.it (G.C.); 7Neurology Unit, Department of Clinical and Experimental Medicine, University Hospital of Pisa, 56124 Pisa, Italy; maestri74@gmail.com (M.M.);; 8Section of Statistics, University Hospital of Pisa, 56124 Pisa, Italy; r.morganti@ao-pisa.toscana.it

**Keywords:** thymectomy, thymic neoplasm, myasthenia gravis, mediastinal surgery, robotic surgery, sternotomy

## Abstract

**Simple Summary:**

Extended thymectomy is the gold standard in the treatment of patients with thymic neoplasm and affected by myasthenia gravis. For a long time, the traditional approach has been sternotomy, though the application of minimally invasive techniques has spread in recent decades. Several authors have demonstrated the safety and feasibility of minimally invasive thymectomy. This multicenter study aims to compare the outcomes of robotic and open thymectomy in myasthenic patients affected by thymic tumors. Short-term and long-term results were presented, showing how the robotic approach can be considered comparable to open surgery in terms of oncological radicality and the improvement of myasthenic symptomatology, with associated faster recovery.

**Abstract:**

Thymectomy is the gold standard in the treatment of thymic neoplasm and plays a key role in the therapeutic path of myasthenia gravis. For years, sternotomy has been the traditional approach for removing anterior mediastinal lesions, although the robotic thymectomy is now widely performed. The literature is still lacking in papers comparing the two approaches and evaluating long-term oncological and neurological outcomes. This study aims to analyze the postoperative results of open and robotic thymectomy for thymic neoplasms in myasthenic patients. Surgical, oncological and neurological data of myasthenic patients affected by thymic neoplasms and surgically treated with extended thymectomy, both with the open and the robotic approach, in six Italian Thoracic Centers between 2011 and 2021 were evaluated. A total of 213 patients were enrolled in the study: 110 (51.6%) were treated with the open approach, and 103 (48.4%) were treated with robotic surgery. The open surgery, compared with the robotic, presented a shorter operating time (*p* < 0.001), a higher number of postoperative complications (*p* = 0.038) and longer postoperative hospitalization (*p* = 0.006). No other differences were observed in terms of surgical, oncological or neurological outcomes. The robotic approach can be considered safe and feasible, comparable to the open technique, in terms of surgical, oncological and neurological outcomes.

## 1. Introduction

Thymic neoplasms are rare tumors. In particular, the most common tumors of the mediastinum are represented by thymomas, accounting for 0.2–1.5% of all malignancies [1,2]. Thymomas are associated in 10–15% of cases with myasthenia gravis (MG), an autoimmune neuromuscular disease that represents the most common paraneoplastic syndrome associated with these neoplasms [2,3,4]. Surgery is the gold standard in the treatment of thymic lesions, including thymoma [5], in which a radical thymectomy with the completeness of resection and negative margins is considered the most important prognostic factor [6,7]. In fact, despite their indolent behavior, thymomas are capable of invading surrounding structures and spreading to pleura and distant organs [2]. In addition, extended thymectomy has been shown to improve myasthenic symptoms, representing an effective procedure in the treatment of both non-thymomatous and thymomaous patients affected by MG [8,9]. The traditional surgical approach for thymectomy has been median sternotomy for a long time [5,10], although the application of minimally invasive surgery (MIS) has increasingly spread in recent decades. The European Society for Medical Oncology (ESMO) guidelines, published in 2015, consider the MIS a valuable option for thymic tumors, following the principle of achieving complete resection [11]. Currently, the use of the minimally invasive technique for thymoma is also considered by the National Comprehensive Cancer Network (NCCN) guidelines [12]. However, the currently available guidelines recommend MIS only in early-stage thymic tumors due to the limited published data on the more advanced stages.

Initially, since the first thoracoscopic thymectomy was performed by Sugarbaker in 1993 [13], most of the concern with the use of minimally invasive surgery was regarding the possible rupture of the capsule with tumor seeding and higher risk of recurrences during endoscopic manipulations of the neoplasm [1,7]. Nevertheless, the robotic approach provides several benefits in the treatment of mediastinal disease, overcoming the conventional thoracoscopy limitations and offering several technical advantages, such as the 10-time-magnified three-dimensional vision, the highly precise movements and the effortless visualization of the operating field. Furthermore, robotic surgery allows the application of the ‘no-touch technique’, according to the International Thymic Malignancy Interest Group (ITMIG) recommendations, and offers lower operative trauma and postoperative morbidity, reduced hospitalization stay and faster postoperative recovery [1,3,7,14]. Several studies have demonstrated the safety and feasibility of robotic thymectomy, also for thymomatous disease, though the number of reports of the oncological and neurological long-term outcomes, also in the advanced stages, in the literature is still low.

This multicenter Italian study primarily aims to compare the surgical, neurological and oncological outcomes of robotic and open thymectomy for thymic neoplasms in patients affected by MG and to validate the safety and feasibility of the robotic approach in advanced stages.

## 2. Materials and Methods

This is a retrospective multicenter study collecting data from all myasthenic patients who underwent robotic and open extended thymectomy for thymic neoplasm from 2011 to 2021 in six Italian Thoracic Surgery Centers (University Hospital of Bari, University Hospital of Bologna, University Hospital of Firenze, University Hospital of Pisa, IRCCS Regina Elena National Cancer Center-Rome, Unicamillus-Saint Camillus University of Health Sciences-Rome). Patients with benign lesions, with non-thymic origin of the cancer or in which a macroscopically radical resection was not achievable (R2) were excluded from this study. Patients who were candidates for debulking (R2) were excluded due to the different outcomes related to unradical resection in thymoma. All patients underwent preoperative neurological evaluations and blood tests to reveal the presence of acetylcholine receptor (AChR) and muscle-specific serum kinase (MuSK) antibodies to confirm the diagnosis of MG. Neurological status was classified according to the Osserman–Jankins classification and Myasthenia Gravis Foundation of America (MGFA) classification. Mediastinal masses were detected by computed tomography (CT) scan or magnetic resonance imaging (MRI) of the thorax.

The open approach adopted was median sternotomy or lateral or posterolateral thoracotomy, while the robotic approach was totally endoscopic with the use of three arms. According to the ITMIG recommendations, an extended thymectomy was always performed, consisting of the complete dissection of the thymic gland en-bloc with the mediastinal adipose tissue between the two phrenic nerves and the innominate vein. Clinical, surgical, neurological and oncological data were retrospectively collected and analyzed. Primarily, the aim of this work was the evaluation of short- and long-term postoperative outcomes (surgical, neurological and oncological). Second endpoint was the validation of the safety and feasibility of the robotic approach in advanced stages. From an oncological point of view, the specimen was histologically classified according to the World Health Organization (WHO) classification of thymic epithelial tumors and the Masaoka–Koga staging system.

Neurological outcomes were obtained by a dedicated neurological team with periodical evaluations and assessed via preoperative and postoperative Myasthenia Gravis Composite (MGC) score, Myasthenia Gravis Activities of Daily Living (MG-ADL) and the Myasthenia Gravis Foundation of America Post-Intervention Score (MGFA-PIS). A complete stable remission (CSR) was defined as no symptoms and without medication for at least 1 year [15]. In addition, steroid and pyridostigmine postoperative reduction was evaluated.

Adjuvant radiotherapy was considered in advanced stages, in case of aggressive histology or R1 resection, according to guidelines [11]. Overall survival is defined as the time from surgical procedure to death from any cause, while disease-free survival refers to the time from operation until the recurrence of disease or death. Oncological follow-up has been performed with periodical CT scans of the thorax every 6 months for 2 years, then annually [12].

### Statistical Analysis

Categorical data were described with absolute and relative (%) frequency, and continuous data were summarized with mean and standard deviation because they were normally distributed. To compare continuous and categorical surgical/oncological outcomes between surgical methods (robot and open), Student’s test for independent samples and chi-square test were performed, respectively. Survival analysis was performed by univariate Cox regression, and survival curves were calculated with the Kaplan–Meier method. Significance was set at 0.05, and all analyses were carried out by SPSS v.28 technology.

## 3. Results

Data from six Italian Thoracic Centers were retrospectively analyzed, evaluating a total of 213 myasthenic patients affected by the thymic tumor; of them, 98 (46%) were males, and 115 (54%) were females, with an average age of 55 (SD ± 14.7) years.

### 3.1. Surgical Results

The sample consisted of two groups based on the surgical approach: the open surgery group, comprising 110 (51.6%) patients (72.7% sternotomy, 27.3% thoracotomy), and the robotic surgery group, which included 103 (48.4%) patients (Figure 1). A conversion from the robotic to the open approach was needed in three (2.9%) advanced-stage cases. No intraoperative complications were observed. The two groups were comparable in terms of demographic characteristics (Table 1).

Postoperative outcomes are described in Table 2. The examination of surgical results showed a shorter operation time in the open approach when compared to the robotic surgery (*p* < 0.001) but a higher rate of postoperative complications (*p* = 0.038), both with statistically significant differences. The main postoperative complications are described in Table 2. The median hospitalization stay was shorter for the robotic group (*p* = 0.006), while the median chest tube drainage duration was similar in both. Moreover, tumors of larger dimensions were treated mainly with the open approach (*p* < 0.001).

### 3.2. Oncological Results

A radical resection of the neoplasm (R0) was obtained in 208 (97.6%) patients of the total sample, with a similar R1 rate when comparing the two groups (*p* = 0.705). The histopathological characteristics and the Masaoka–Koga staging are described in Table 3. The advanced stages included in this study presented macroscopic invasion of neighbor organs (lung, pericardium) or pleural dissemination; nevertheless, a vascular invasion was never reported.

Adjuvant radiotherapy was administered more frequently in patients who underwent open surgery (*p* < 0.001) (Table 3).

The mean oncological and neurological follow-up was 58 (SD ± 36) for the total sample, being longer for the open group: 78 months (SD ± 32) versus 39 months (SD ± 28) (*p* < 0.001). The OS of the total sample was 94% at 3 years and 86% at 5 years, while DFS was 98% at 3 years and 96% at 5 years. No significant differences resulted between the two groups in terms of recurrence (*p* = 0.209), DFS (*p* = 0.937) and OS (*p* = 0.701) (Table 3) (Figure 2).

### 3.3. Neurological Results

Neurological outcomes are described in Table 4. No statistically significant differences were observed comparing the robotic and the open approach in the evaluation of postoperative reduction of MG symptoms (*p* = 0.231), postoperative reduction of MGC score (*p* = 0.171), postoperative reduction of MG-ADL (*p* = 0.066) (Figure 3) and postoperative reduction of pyridostigmine (*p* = 0.135). Conversely, the postoperative reduction of steroids had inferior results in patients treated with robotic surgery (*p* < 0.001). The histopathological WHO category did not influence the postoperative reduction of MG symptoms (*p* = 0.237).

## 4. Discussion

Thymectomy is a surgical procedure indicated in the treatment of lesions of the anterior mediastinum, in particular, representing the gold standard in the case of resectable thymic tumors [14,16,17]. Furthermore, thymectomy represents a valid therapeutic option for patients with a diagnosis of MG, being the best therapeutic choice in generalized disease and in the presence of a particular antibody pattern consisting of seropositive acetylcholine receptor antibodies and seronegative muscle-specific kinase protein [3,17,18,19].

Weigert and Bell first noted the prevalence of thymic abnormalities, such as hyperplasia and tumors, during the autopsy of patients affected by MG. On this finding, in 1939, Blalock presented the first series of patients affected by thymoma and MG surgically treated by transsternal thymectomy [14,20,21].

When associated with thymoma, MG seems to be more severe, and the postoperative results are not as good as in those patients with non-thymomatous disease. Nevertheless, when feasible, surgery should never be delayed in patients affected by thymoma, adjuvanted by radiotherapy or medical therapy when not radical, both for the control of myasthenic symptoms and for the prevention of neoplastic local spreading [22,23]. An extended thymectomy, which means the complete resection of the thymic gland en-bloc with the perithymic fat tissue, is recommended to optimize the results. In detail, the ITMIG guidelines thus recommend the removal of the tumor with the thymic gland and the anterior mediastinal fat tissue after the identification of both the right and left phrenic nerves, proceeding with the dissection of tissue from the jugular to the anterior pericardiophrenic angle, to ensure oncological and neurological radicality [9,14].

Transsternal thymectomy has been considered the gold standard surgical approach for a long time due to the wide operating field and the better visualization of both phrenic nerves and vascular structures. Sternotomy, indeed, enables a safer dissection in complex cases with infiltration of neighbor structures [14,16]. Nevertheless, the open approach is characterized by high invasiveness, with a higher risk of bleeding and wound infection, great postoperative pain and longer recovery, especially in the elderly. Over the years, alternative approaches were proposed for thymectomy, such as transcervical thymectomy, thoracoscopic unilateral or bilateral thymectomy and robot-assisted thymectomy [16]. This trend has been toward increasingly less invasive approaches to reduce surgical trauma and to obtain a faster postoperative recovery. Since the first thoracoscopic thymectomy performed by Sugarbaker in 1993 [13] and the first robotic thymectomy by Yoshino in 2000 [24], several studies have shown the safety and feasibility of minimally invasive thymectomy, both for video-thoracoscopic (VATS) and robot-assisted thymectomy [14,16,21]. To date, the minimally invasive approach is recommended by the NCCN and the ESMO guidelines in the early stages, provided that radical resection is feasible and performed by expert surgeons [11,12]. Minimally invasive thymectomy offers decreased operative blood loss, lower hospital length of stay, reduced operative trauma, less postoperative morbidity and faster postoperative recovery [1,3,7,14]. Robotic thymectomy overcomes the limits of VATS with the highly precise movements, the 7 degrees of freedom of the endowrists and the effortless visualization of the operating field thanks to the high definition and three-dimensional view [14,16]. Furthermore, robotic thymectomy more perfectly adapts to different anatomical variations when compared with other minimally invasive techniques, permitting the achievement of all the principles of the ITMIG recommendations [17]. The results reported from previous papers have allowed us to overcome the concern about the risk of rupture of the tumor capsule, with tumor seeding and a higher risk of recurrences, during endoscopic manipulations of the neoplasm [1,7].

In 2012, Marulli et al. published the first multicenter retrospective study, describing the outcomes of 74 patients who underwent robotic thymectomy for early-stage thymoma, confirming the robotic approach as safe and feasible, with encouraging oncological outcomes [25]. These results were confirmed by the same authors in 2016, in a multicenter study with a larger sample of patients, by extending the indication of robotic thymectomy also to selected cases of advanced stages [6].

Some studies have compared the open with minimally invasive thymectomy, even though the results on long-term oncological and neurological outcomes are still poor. Jurado et al., in a monocentric study, compared MIS (75 thoracoscopic and 2 robotic thymectomies) and open thymectomy (186 patients) in a cohort of 263 patients treated for benign and malignant mediastinal lesions. The minimally invasive approach presented shorter hospital stays, shorter intensive care unit length of stay, lower estimated blood loss and lower complication rates, with comparable operative time, radicality of resections and neurological outcomes [5].

Marulli et al. analyzed data from 164 patients with early-stage thymoma, confirming that the robotic approach, compared with median sternotomy, presented lower intraoperative blood loss, less perioperative complications, shorter time to chest drainage removal and hospital discharge, but longer operative time [1]. In 2019, Yang et al., in their national analysis of outcomes of open and robotic thymectomy for stage I-III thymoma, confirmed that robotic surgery is safe and feasible without compromising oncologic efficacy [26]. Similar results were observed in a systematic review by Buentzel et al., showing several advantages in patients who underwent robotic surgery [27]. Another systematic review by O’Sullivan et al., comparing VATS, robotic and open approaches, confirms positive results in favor of robotic surgery [28].

Recently, Lee et al. published the first systematic review and meta-analysis evaluating long-term outcomes in the comparison between MIS and open thymectomy in myasthenic patients, reporting advantages of MIS in postoperative results [29]. Nevertheless, this study is the first multicenter comparative study between open and robotic surgery in myasthenic patients with thymic tumors, presenting surgical, oncological and neurological outcomes. In line with the literature, robotic thymectomy is associated with lower postoperative complication rate and hospital stay despite longer operative time when compared to open surgery. The increased duration of the robotic operation is directly related to the features of the robotic platform, with its docking/undocking time; this factor seems to be negligible, given the favorable postoperative recovery, with the related cost and social benefits obtained in patients treated by robotic surgery. Chylothorax was observed in three cases in the robotic group. This incidence is indeed in line with the literature, although chylothorax is a rare postoperative complication that can occur after radical thymectomy, especially in myasthenic patients, as reported by Zhang et al. [30].

The analysis of the oncological outcomes in our study found that open and robotic surgery are associated with similar results in terms of surgical radicality DFS and OS. Adjuvant radiotherapy was mostly performed in the open group; these data may be related to the larger presence of advanced stages in this population. In our study, recurrences occurred in 5.5% of patients treated by open thymectomy and in 2.9% of the patients who underwent robotic thymectomy: the difference observed can be related to the major presence of Masoka III-IV stages (14.5% versus 7.8%) and to the longer follow-up time in the open group. A lower rate of recurrences was similarly observed by Marulli et al. in their multicenter study analyzing the outcomes of early-stage thymoma, with only one (0.7%) patient with pleural recurrence and a 100% thymoma-related survival [6]. Additionally, Casiraghi et al. found only one case (2.1%) of recurrence in the open group while comparing open and robotic surgery in patients with early-stage thymoma (25 open patients vs. 25 robotic) [7]. A similar 5-year OS was also observed by Yang et al. in the comparison between open and robotic thymectomy for the treatment of thymoma [26]. In addition, a large cohort study based on the national cancer database conducted by Salfity et al., comparing VATS (305 cases), robotic (342 cases) and open (1665 cases) thymectomy, showed encouraging oncological outcomes associated with MIS procedures. The authors indeed noticed a higher radical resection rate and better survival in the robotic group [31].

In the evaluation of surgical results, the bigger neoplastic lesions that emerged were preferably treated with open surgery, especially in the first years taken into account. In fact, the management of large mediastinal neoplasms is more challenging, requiring greater surgical expertise and accuracy in movements to respect the “no-touch technique” and to avoid tumors spreading. Kneuertz et al. confirmed the feasibility of robotic surgery in the treatment of large thymoma, obtaining the same radicality (R0) and complication rate when comparing robotic surgery and sternotomy [10]. In our experience, as surgeons have become more experienced over the years, we have seen an increase in the size of thymomas treated with the robotic approach. Moreover, our study confirms the safety of robotic surgery in the treatment of advanced-stage thymic tumors, showing no difference in terms of postoperative complication rate, radicality and oncological and neurological outcomes when compared to the open approach.

In our series, neurological symptoms resulted in an improvement in 70% of the patients treated by open surgery and in 67% of the patients who underwent robotic thymectomy. Furthermore, a complete remission rate (CSR) was observed in 12 (10.9%) and 5 (4.9%) patients, respectively, with no significant difference. These findings were in line with Jurado et al. results, which compared open and minimally invasive thymectomy, obtaining an improvement in myasthenic symptoms in 70% of the MIS group and 67% of the open group [5]. A neurological benefit with a reduced dose of steroid assumption was also observed by Keijzers et al. [18]. They analyzed data from 125 myasthenic patients who underwent robotic thymectomy; 77% of them showed neurological improvement, with a three-year probability of CSR of 28.2%. In our study, the robotic group presented less postoperative reduction of steroids; these data are probably related to the differences in follow-up between the two groups, with the patients treated with open thymectomy having longer follow-ups (78 months ± 32 versus 39 months ± 28). According to Aydin et al., in fact, remission is not immediate after thymectomy, consisting of less than 20% in the first year, increasing to 50% over 7–10 years [23]. Moreover, our findings are in line with the monocenter study conducted at our clinic on patients affected by MG who underwent robotic surgery for thymomas by Romano et al. in 2021. The authors reported an improvement in symptoms and drug assumption after robotic thymectomy, with a CSR of 14.7%, confirming that the robotic approach is also safe from a neurological point of view [14].

### Limitations

The limits of this study are its retrospective nature and the differences in follow-up duration between the two groups evaluated. In particular, the limitations of this retrospective study are represented by the presence of incomplete data and the selective bias due to the non-randomized allocation of patients to the two groups. In addition, being a multicenter study, there is no certain uniformity in the preoperative neurological management of patients. Further prospective studies should be conducted to compare minimally invasive and open techniques to more properly analyze neurological and oncological outcomes, focusing on advanced stages of thymic tumors.

## 5. Conclusions

This study represents the first multicenter study comparing short-term and long-term results of robotic and open thymectomy for the treatment of thymic neoplasm in myasthenic patients. The statistical analysis showed lower hospitalization stays and postoperative complication rates in patients who underwent robotic thymectomy despite a longer operative time, with no difference regarding neurological and oncological outcomes.

Robotic thymectomy can be considered a safe procedure, comparable to open thymectomy in terms of oncological and neurological results, even in more advanced tumors.

## Figures and Tables

**Figure 1 cancers-16-00406-f001:**
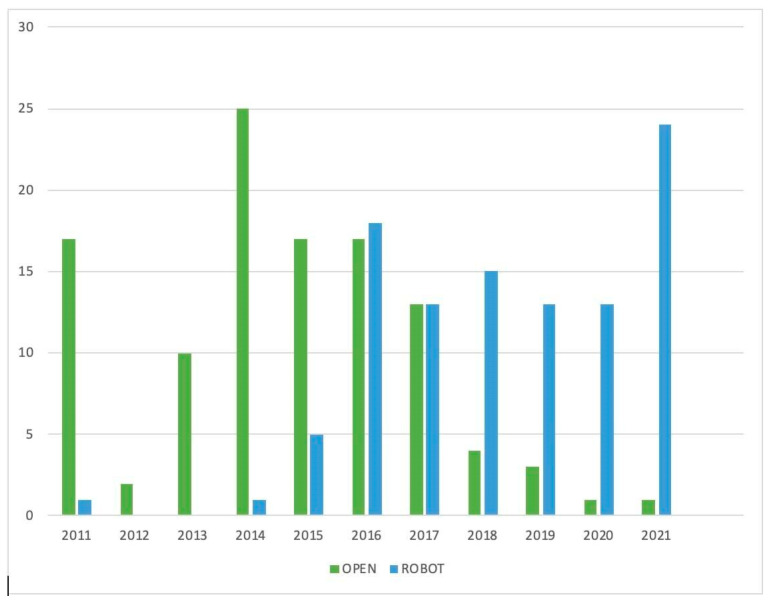
Distribution of surgical procedures over the years.

**Figure 2 cancers-16-00406-f002:**
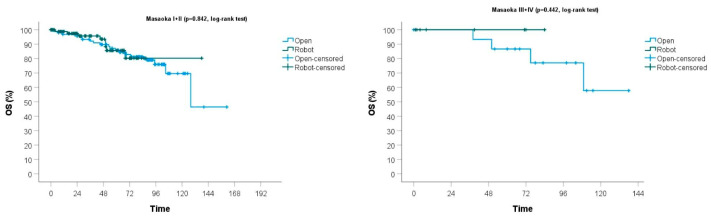
Overall survival is stratified according to the Masaoka–Koga stages.

**Figure 3 cancers-16-00406-f003:**
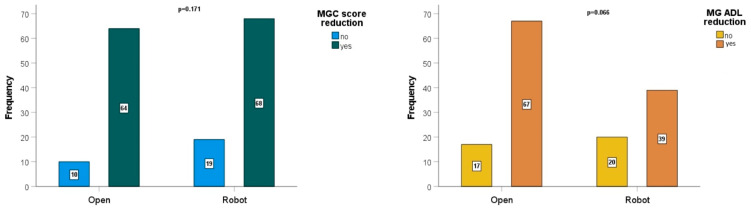
MGC score and MG-ADL score reduction.

**Table 1 cancers-16-00406-t001:** Demographic characteristics.

	All Patients *n* = 213	Open Surgery *n* = 110 (51.6%)	Robotic Surgery *n* = 103 (48.4%)
Male	98 (46%)	50 (45.5%)	48 (46.6%)
Female	115 (54%)	60 (54.5%)	55 (53.4%)
Age (years)	55 (14.7)	55.2 (13.9)	55 (15.6)
Steroid at operation time (mg)	32.8 (22.3)	34.1 (20.5)	31.3 (24.2)
Pyridostigmine at operation time (mg)	134.1 (101.1)	141.5 (97.6)	125.8 (104.7)

Statistics: mean (SD) or frequency (%).

**Table 2 cancers-16-00406-t002:** Postoperative outcomes and postoperative complications.

	All Patients *n* = 213	Open Surgery *n* = 110 (51.6%)	Robotic Surgery *n* = 103 (48.4%)	*p*-Value
Operative time (minutes)	127.8 (57.1)	110 (42)	146 (62)	<0.001
Extubation day				
OP	206 (96.7%)	104 (94.5%)	102 (99%)	0.578
I POD	3 (1.4%)	2 (1.8%)	1 (1%)	
Complications	37 (17.4%)	22 (20%)	15 (14.6%)	0.038
Need of transfusion	5 (2.3%)	4 (3.6%)	1 (1%)	0.199
Hospitalization days	6.5 (5.5)	7.5 (6)	5.5 (4.8)	0.006
Chest tube days	3.4 (3.5)	3.3 (2)	3.5 (4.6)	0.634
Postoperative complications				
Anemia requiring	5 (2.3%)	4 (3.6%)	1 (1%)	
blood transfusion				
Arrhythmias	8 (3.8%)	5 (4.5%)	3 (2.9%)	
Chylothorax	3 (1.4%)	0	3 (2.9%)	
Exacerbation/myasthenic crisis	6 (2.8%)	3 (2.7%)	3 (2.9%)	
Neuromyotonia	1 (0.5%)	1 (0.9%)	0	
Thoracic complications (pneumothorax, pleural effusion)	6 (2.8%)	5 (4.5%)	1 (1%)	
Myocardial infarction	1 (0.5%)	1 (0.9%)	0	

OP: operative day; I POD: first postoperative day. Statistics: mean (SD) or frequency (%).

**Table 3 cancers-16-00406-t003:** Histopathological classification.

	All Patients *n* = 213	Open Surgery *n* = 110 (51.6%)	Robotic Surgery *n* = 103 (48.4%)	*p*-Value
Masaoka–Koga stage				
I	58 (27.2%)	12 (10.9%)	46 (44.7%)	0.604
II	131 (61.5%)	82 (74.6%)	49 (47.5%)	
III	20 (9.4%)	13 (11.8%)	7 (6.8%)	
IV	4 (1.9%)	3 (2.7%)	1 (1%)	
WHO				
A	31 (14.6%)	20 (18.2%)	11 (10.7%)	0.075
AB	36 (16.9%)	21 (19.1%)	15 (14.6%)	
B1	34 (15.9%)	12 (10.2%)	22 (21.4%)	
B2	85 (39.9%)	47 (42.8%)	38 (36.9%)	
B3	12 (5.6%)	8 (7.3%)	4 (3.8%)	
C	2 (1%)	0	2 (1.8%)	
Others	13 (6.1%)	2 (1.8%)	11 (10.7%)	
Tumor size (cm)	5.2 (2.6)	5.6 (2.6)	3.5 (2.1)	<0.001
R0	208 (97.6%)	107 (97.3%)	101 (98%)	0.705
R1	5 (2.4%)	3 (2.7%)	2 (2%)	
Postoperative Radiotherapy	72 (33.8%)	51 (46.4%)	21 (20.4%)	<0.001
3-year OS	94%	93%	95%	
5-year OS	86%	86%	87%	

Statistics: mean (SD) or frequency (%).

**Table 4 cancers-16-00406-t004:** Neurological outcomes.

	All Patients *n* = 213	Open Surgery *n* = 110 (51.6%)	Robotic Surgery *n* = 103 (48.4%)
MFGA-PIS			
CSR	17 (8%)	12 (10.9%)	5 (4.9%)
MM	73 (34.3%)	47 (42.7%)	26 (25.2%)
PR	2 (0.9%)	1 (0.9%)	1 (0.9%)
I	93 (43.7%)	63 (57.3%)	30 (29.1%)
U	30 (14.1%)	16 (14.5%)	14 (13.6%)
W	14 (6.6%)	3 (2.7%)	11 (10.7%)
E	3 (1.4%)	2 (1.8%)	1 (0.9%)
Preoperative MGC score	7.1 (8.1)	7 (8.2)	7.1 (8.1)
Postoperative MGC score	2.8 (5.5)	2.8 (5.6)	2.8 (5.5)
Preoperative MG-ADL	3.8 (3.3)	3.9 (3.3)	3.8 (3.3)
Median postoperative			
MG-ADL	1.8 (2.8)	1.8 (2.9)	1.8 (2.8)
Postoperative reduction of steroid	117 (54.9%)	70 (63.6%)	47 (45.6%)
Postoperative reduction of pyridostigmine	91 (42.7%)	53 (48.2%)	38 (36.9%)
Postoperative reduction of MG symptoms	146 (68.5%)	77 (70%)	69 (67%)

MGFA-PIS: Myasthenia Gravis Foundation of America Post-Intervention Score; MGC score: Myasthenia Gravis Composite score; MG-ADL: Myasthenia Gravis Activities of Daily Living; CSR: complete symptom remission; MM: minimal manifestation; PR: pharmacologic remission; I: improved; U: unchanged; W: worse; E: exacerbation. Statistics: mean (SD) or frequency (%).

## Data Availability

The data underlying this article will be shared by the corresponding author upon reasonable request.

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
