# Peer review of "National Multicenter Study on the Comparison of Robotic and Open Thymectomy for Thymic Neoplasms in Myasthenic Patients: Surgical, Neurological and Oncological Outcomes"

_cancers, 2024, doi:10.3390/cancers16020406_

Round 1
Reviewer 1 Report
Comments and Suggestions for Authors
Congratulations on your work. It is well written and I enjoyed reading it.
1. Why were pts where a macroscopically radical resection was not achievable (R2) excluded from the study? Were these more common in one of the groups.
2. Do the converted patients sit in the open or robotic group? It is not made clear in the paper. It is probably best done as an intention to treat ie sit in robotic group.
3.Where appropriate in you tables can you include a range?
4. Were any procedures performed VATS during this time?
5. Does the diameter of tumour operated on robotically change over time? At a guess it gets larger.
6. You said the limitations on this study is its retrospective nature. Please can you ellaborate why this may have biased the results in a bit more detail
Author Response
We would like to thank you for your precious comments and suggestions.
- Patients candidate to debulking (R2) were excluded due to the different outcomes related to unradical resection in thymoma. No R2 were obtained in patients in which radical resection was planned. (Line 105-106)
- Patients who underwent conversion to open surgery were patient treated with robotic surgery.
- The data in the table are expressed as a mean, so in line with our statistics we have preferred to report the standard deviation rather than the range.
- The study is a multicenter experience, so there is a possibility that VATS thymectomy was performed in some centers. However, VATS thymectomy was not the focus of this work and that's why we didn't report these patients.
- In our experience, as surgeons have become more experienced over the years, we have seen an increase in the size of thymomas treated with the robotic approach. (Line 322-324)
- As mentioned above, one of the major limitations of our work is its retrospective nature. In particular, the limitations of this retrospective study are the presence of incomplete data and the selective bias due to the non-randomised allocation of patients to the two groups. (Line 351-353)
Reviewer 2 Report
Comments and Suggestions for Authors
I would like to congratulate the authors of an interesting article titled: “National multicenter experience on comparison of robotic and open thymectomy for thymic neoplasms in myasthenic patients: surgical, neurological and oncological outcomes.”
The study conducted by the authors is innovative and the results are very interesting for thoracic surgeons involved in the oncological treatment of anterior mediastinal tumors. The quality of the English is good. The study falls in the scope of this Special Issue of the Cancers.
I have no comments to Simple summary, Abstract, Introduction and Discussion. Simple summary and abstract are well written. Introduction provides sufficient background for the study. Discussion is well written, authors refer all findings to the current literature.
However, some changes could be introduced to improve the methodology of the study and results presentation.
1. Material and methods are clearly written, but in my opinion require changes:
a. Endpoints should be clearly specified: one, maximum two primary endpoints (for example complications and 5-year survival), and two – three secondary endpoints.
b. The study may be biased regarding short- and long-term outcomes.
Authors did not provide information on comorbidities and paraneoplastic syndromes, which may have influenced postoperative complications.
Furthermore, the groups differ regarding Masaoka-Koga classification (more stage I patients in robotic group and more stage II and III patients in the open group), which could have influenced survival. Measures to reduce bias could be used - to obtain two similar groups for survival (OS, DFS) analysis, I suggest propensity score matching, including for example variables such as age, comorbidities, Masaoka-Koga, WHO, R and radiotherapy.
2. Results require changes:
a. Table 1 should additionally include information on comorbidities in both groups, including most important diseases known as surgical risk factors (for example chronic obstructive pulmonary disease, coronary arterial disease, cerebrovascular disease, peripheral arterial disease, hypertension, diabetes mellitus, chronic kidney disease) and paraneoplastic syndromes.
b. I suggest moving the data regarding the postoperative course from Table 1 in Table 2 (extubation day, complications, need of transfusion, hospitalization days, chest tube days).
c. Table 3. Listing 13 types of WHO classifications in the table significantly impairs its clarity. WHO classification lists five types of thymoma: A, AB, B1, B2 and B3. As I am aware, type was replaced by thymic carcinoma some time ago. According to the WHO classification, I suggest limiting to five types of thymoma: A, AB, B1, B2 and B3, and replacing "C" and "C/B3" with the phrase "thymic carcinoma".
d. Table 3. P-values for Masaoka-Koga and WHO classifications should be included.
e. Table 4 would be clearer if the abbreviations CSR, MM, PR, I, U, W, E were replaced with full phrases, for example CSR - complete stable remission. Please include p-values in Table 4.
f. All Tables: all abbreviations used in tables should be explained in the footnotes (Table 1: OP, IPOD; Table 4 MFGA-PIS, MGC, MG-ADL, MG).
3. Limitations should be expanded, please consult the literature on limitations of retrospective, multicenter trials. Please also mention the strengths of the study.
4. References should be formatted according to the Cancers guidelines.
Otherwise, I have no comments. Once again, I congratulate the authors on an interesting study.
Comments on the Quality of English Language
Quality of English is good.
Author Response
We would like to thank you for your valuable suggestions and comments, which helped us to improve the manuscript.
- Primarily, the aim of this work was the evaluation of short- and long-term postoperative outcomes (surgical, neurological and oncological). Second endpoint was the validation of the safety and feasibility of the robotic approach in advanced stages. (Line 119-121)
- The study is a multicenter experience, therefore was not possible to obtain information about comorbidities of the whole sample. Survival analysis were stratified according the Masaoka-Koga stage and no differences were observed.
- a. The study is a multicenter experience, therefore was not possible to obtain information about comorbidities of the whole sample. b-c-d-f. We thank you for your suggestion and will change the text accordingly. e. Abbreviations are explained in the footnote. The p-value required is that related to the reduction in myasthenic symptoms and is given in the text. (Line 200)
- As mentioned above, one of the major limitations of our work is its retrospective nature. In particular, the limitations of this retrospective study are the presence of incomplete data and the selective bias due to the non-randomised allocation of patients to the two groups. (Line 351-353) The strengths of the study, as stated in the text, are that it is the first multicentre study to report oncological and neurological long-term outcomes in a large sample of myasthenic patients.
- We thank you for your suggestion and will change the text accordingly.
Reviewer 3 Report
Comments and Suggestions for Authors
Dear Authors:
The authors have carried out a retrospective study titled “National multicenter experience on comparison of robotic and open thymectomy for thymic neoplasms in myasthenic patients: surgical, neurological and oncological outcomes”. The aim of this study was to analyze the postoperative results of open and robotic thymectomy comparing the surgical, neurological and oncological outcomes in myasthenic patients with thymic neoplasms and to validate the safety and feasibility of the robotic approach in advanced stages.
Some considerations need to be considered:
- Although authors refer in the manuscript that the median chest tube drainage duration was similar in both groups (open and robotic) it is showed in table 1 a slighter increase in the robotic group. What do authors attribute it to? This point should be addressed in the discussion section.
- Adjuvant radiotherapy was administered more frequently in patients who underwent open surgery. This fact should be more detailed developed in the discussion section because it is relevant.
- The presence of chylothorax as a postoperative complication in robotic surgery is greater and this fact should be briefly discussed in the manuscript given that minimally invasive surgery usually involves less tissue manipulation.
- Have the authors evaluated the surgeon variable in this study? Experience and surgical casuistry may represent a bias in the results when evaluating both surgical approaches.
- I strongly agree with the limitations described by the authors that could affect the results of the study.
Kind regards
Author Response
Dear Authors:
The authors have carried out a retrospective study titled “National multicenter experience on comparison of robotic and open thymectomy for thymic neoplasms in myasthenic patients: surgical, neurological and oncological outcomes”. The aim of this study was to analyze the postoperative results of open and robotic thymectomy comparing the surgical, neurological and oncological outcomes in myasthenic patients with thymic neoplasms and to validate the safety and feasibility of the robotic approach in advanced stages.
Some considerations need to be considered:
- Although authors refer in the manuscript that the median chest tube drainage duration was similar in both groups (open and robotic) it is showed in table 1 a slighter increase in the robotic group. What do authors attribute it to? This point should be addressed in the discussion section.
- Adjuvant radiotherapy was administered more frequently in patients who underwent open surgery. This fact should be more detailed developed in the discussion section because it is relevant.
- The presence of chylothorax as a postoperative complication in robotic surgery is greater and this fact should be briefly discussed in the manuscript given that minimally invasive surgery usually involves less tissue manipulation.
- Have the authors evaluated the surgeon variable in this study? Experience and surgical casuistry may represent a bias in the results when evaluating both surgical approaches.
- I strongly agree with the limitations described by the authors that could affect the results of the study.
Kind regards
We would like to thank you for the revision and the good points you have brought to our attention.
- There is no statistical difference in chest tube duration between the two groups, and the mean difference is very small. The duration of the chest tube is slightly longer in the robotic group because the trial includes surgery with the first experience of the robotic approach. With increasing experience, the chest tube duration also decreased. In fact, the duration of chest tube is shorter in centres with more experience and a higher number of cases.
- Adjuvant radiotherapy was mostly performed in the open group; these data may be related to the larger presence of advanced stages in this population. (Line 304-305)
- Chylothorax was observed in three cases only in the robotic patients. This incidence is indeed in line with the literature, although chylothorax is a rare postoperative complication that can occur after radical thymectomy, especially in myasthenic patients, as reported by Zhang et al. Further studies would be necessary to understand risk factors and possible causes. (Line 298-301)
Zhang Z, Hu X, Chen Q, Li H. Chylothorax after thoracoscopic extended thymectomy: a case report and literatures review. J Thorac Dis. 2018 Aug;10(8):E639-E642.
- This is a multicentre trial. All centres involved are training centres, so some operations were performed by young surgeons. A prospective study would be needed to better assess surgeon variability.